# Investigating the Effects of Non-Branded Foods Placed in Cartoons on Children’s Food Choices through Type of Food, Modality and Age

**DOI:** 10.3390/ijerph16245032

**Published:** 2019-12-10

**Authors:** Victoria Villegas-Navas, Maria-Jose Montero-Simo, Rafael A. Araque-Padilla

**Affiliations:** Department of Management; Universidad Loyola Andalucia, Escritor Castilla Aguayo St., 4, 14004 Cordoba, Spain; jmontero@uloyola.es (M.-J.M.-S.); raraque@uloyola.es (R.A.A.-P.)

**Keywords:** non-branded food placements, children, low nutritional value foods, cartoons, age differences

## Abstract

Cartoons are among the most consumed media products by children, especially those at a young age. While branded food placements are not allowed in animated series, non-branded food placements are prevalent. However, little is known about the effects that these food placements might have on children’s eating preferences. In an experimental study with 124 children (51.6% girls, age range: 7–11, M_age_ = 9.24, and SD = 1.19), 62 children in the experimental condition were exposed to 16 food placements in cartoons, whereas children in the control condition were exposed to cartoon scenes without foods. The healthiness of the placed foods (low nutritional value foods versus high nutritional value foods) as well as the modality of food placements (unimodal versus bimodal) were manipulated. After watching the cartoon scenes, children completed a choice task where each placed food appeared on a separate choice card. Our results indicate that non-branded low nutritional value foods placed in cartoons are an effective strategy in modifying children’s food choices when children are under age 9. We suggest that policy makers, particularly those involved in the content design of cartoons, take these results into account when placing low nutritional value foods in cartoons, especially for an animated series that targets young child audiences.

## 1. Introduction

At present, children spend a large portion of their leisure time in front of screens [1]. Watching television, playing videogames and surfing the net are part of children’s daily activities. Previous content analyses have shown that while they are consuming different media, children receive a variety of food messages that appear in various formats. Advertisements [2], advergames [3] and food placements [4,5,6] are some of the communication tools that children are exposed to in the media environment. Regarding the latter, the study of food placements targeting children has acquired major relevance in the last decade since Auty and Lewis (2004) [7] found the short-term effectiveness of these portrayals on children’s behavior.

Food placements are typically portrayed within the scenes of entertainment media as product placements [8]. This communication tool is considered to cause less interference than food commercial ads because food placements do not interrupt the storyline of the corresponding entertainment media. In the digital era, where food ads can be skipped by audiences, food placements constitute a key communication strategy when transmitting messages about food patterns.

Food placements might be associated with the problem of childhood overweight and obesity. Since previous studies found that low nutritional value food placements are effective in changing children’s behavior towards foods high in fat, sugar or salt [8,9,10,11], special attention must be paid to this strategy when addressing the problem of childhood obesity. Overweight and obese children are increasingly reaching higher figures worldwide [12,13]. The problem compromises not only today’s children but also tomorrow’s adults, as overweight children are more likely to become overweight adults [14] who experience a greater propensity to develop cardiovascular diseases as well as diabetes at a young age [13]. Children’s self-esteem and confidence are also affected by this problem [15]. These are some of the reasons why The World Health Organization (WHO, 2019) [13] poses the problem of childhood obesity as among the most serious public health challenges of the 21st century.

To date, most of the studies about food placements directed at children have focused on branded foods [7,9,10,11]. While branded food placements are frequent in movies [4] and advergames [3] targeting children, non-branded food placements are the type of products found in cartoons [5], as branded food placements are prohibited in these animated series [16]. This prohibition aims to protect infants against the persuasive effects of branded product placements as children constitute a vulnerable target group [17].

Additionally, although the studies that analyze the effects of food placements are mostly about branded foods, several studies about non-branded food placements can also be found. To our knowledge, approximately half of the studies about the effectiveness of non-branded food placements have focused on foods placed in videogames [18,19,20,21], whereas the other half have focused on video format, such as cartoons or TV shows [8,22,23,24]. Regarding the studies that analyze non-branded foods placed in cartoons, we only found one study that investigated the influence of type of food on children’s behavior [8]. However, that study [8] used Powtoon software to create the animated cartoons, while, in the present study, the animated cartoons were those broadcasted in contemporary children’s TV channels.

In recent decades, despite the large number of studies that have analyzed the effects of branded food placements portrayed in video format [7,9,10,11,25,26,27,28], it is necessary to inquire whether there are also effects in the case of non-branded food placements, which is the objective of the present study. In studies about branded food placement effectiveness, generally, children have to choose between competitors’ brands—that is, products that belong to the same food category, such as soft drinks [7], snacks [10,11] or chocolates [26]. However, in studies about non-branded food placements, the options are not limited to a single food category; in contrast, the range of options is more diverse. This intrinsic difference between the nature of branded and non-branded food placement studies that measure choice leads us to question whether the effects of branded food placements might be extrapolated to non-branded food placements. Additionally, children might see branded foods as a combination of attributes, benefits and perceptions [29] in a different manner than non-branded foods. Overall, marketing campaigns are usually more focused on reinforcing and extending the image of branded foods compared to non-branded foods and this difference might result in distinct affective evaluations by the consumers [30]. Last, persuasive knowledge about non-branded foods might not be activated with the same intensity as with branded foods, especially among young children [31].

Cartoons are among the most consumed media products by children [32]. Children spend a large amount of time watching these animated series [33]. It is not surprising that cartoon characters might constitute a powerful source of influence with regard to eating patterns. Dora the Explorer and fruits, SpongeBob SquarePants and hamburgers (Krabby Patty) or Michelangelo (from Teenage Mutant Ninja Turtles) and pizzas are some of the long line of examples of cartoon characters associated with specific food placements. Each day, children watch the cartoons they are interested in (provided that parents agree, and cartoons are available for watching), and each day, children develop parasocial relationships with cartoon characters [34].

Animated cartoons are part of the long list of examples of environmental determinants that might influence children’s food behavior [35]. Cartoon characters might be perceived as models, and food placements might represent objects with meaning. In this context, vicarious learning might occur for children while they are watching their selected cartoon series, or afterwards. According to social cognitive theory (SCT) [35], three main determinants interact reciprocally and influence each other: personal, behavioral and environmental determinants. Thus, children might have certain personal determinants (cognitive, affective and biological factors) that are translated into a major or minor preference for a certain type of food. On the other hand, children might act (behavioral determinants) in a specific way with regard to that type of food (selecting or not selecting the food or consuming the food in a major or minor quantity). Last, children might receive external influences from the environment (foods placed in cartoons) that might produce an acceptance or rejection of that type of food. It is important to highlight that SCT [35] establishes that children’s behavior is not only shaped by personal and environmental factors but also shaped by people’s capability of acting in a way that is self-organized, proactive, self-reflected and self-regulated. Therefore, despite the factors that might affect children’s eating patterns, children might have the possibility of redirecting those influences.

### 1.1. Objectives and Hypotheses

This study aims to analyze the effects of non-branded foods placed in scenes of real cartoons on choice, as this behavioral variable constitutes the first step prior to consuming a product [36]. For that purpose, this study intends to explore how the type of food and placement modality influence non-branded food placement effectiveness. Additionally, children’s age is studied as a potential moderating factor.

#### 1.1.1. Type of Food and Modality

When studying non-branded food placements, the healthiness of the food needs to be considered, as previous experimental studies show that this distinction makes a difference in the effectiveness of non-branded foods placed in entertainment media [8,18,19,21]. For example, in a study by Dias and Agante (2011) [18], significant effects were found of the type of food game on choice (children who played the “less healthy game” chose significantly more “unhealthy foods” than those children who played the “healthy” game). Similar results were found in the experimental designs of Esmaeilpour (2017) [19] and Pempek (2009) [21] that used videogames. However, the aforementioned studies did not use a game condition without foods; therefore, we were unable to determine the true effect that placing foods in games has on children’s food choices.

Overall, evidence suggests that low nutritional value food placements have effects on children’s behavior, both in the case of branded [9,10,11,37] and non-branded [8] placements. However, we did not find previous studies of branded high nutritional value foods placed in video format, as this type of food is usually presented as non-branded. Findings about non-branded high nutritional value food placements are not conclusive: although some studies found effects on children’s choices [24] and intake [20], most of the previous literature did not [8,21,23]. In a recent study by Binder, Naderer, and Matthes (2019) [38], it was found that a non-branded high nutritional value food (raspberries) did not trigger effects on behavioral responses in children aged 6–10 years. Similarly, in another study that used mandarins as a prominent non-branded high nutritional value food placement [8], negative results were found: the placement of mandarins elicited an adverse effect (the significant choice effects of low nutritional value foods) among children aged 6–11 years. Therefore, it seems that no immediate behavioral responses can be triggered when using high nutritional value food placements. Therefore, the first hypothesis of the present study considers type of food as a potential factor that influences children’s food choices:

**H1.** 
*Low nutritional value as opposed to high nutritional value food placements will lead to an increase in food choice.*


The effectiveness of the food placement strategy is affected not only by the type of food but also by the execution factors, which are considered in the literature as variables involved in the effectiveness of product placement [39]. These factors refer to information about how products appear, including modality, level of plot connection [11,40], frequency of product appearance [10], level of character-product interaction [26], and others.

In the literature about product placements, modality is among the most studied execution factors [22,41,42,43]. According to the sensory channel that is used to receive the placement message, three types of product placement can be distinguished: visual or only-seen placements, verbal or only-heard placements and audiovisual placements [44]. Whereas both visual and verbal placements belong to the unimodal category, audiovisual placements are considered bimodal placements. As animated cartoons constitute a passive entertainment media (in contrast with videogames that are active entertainment media) and non-branded food placements do not trigger children’s persuasive knowledge with the same intensity as branded food placements, non-branded bimodal placements might be more effective in changing eating behaviors (choice, willingness to buy or consumption) than unimodal placements [22]. Further, bimodal placements are considered as more salient (in terms of prominence) than unimodal placements [5,22] and SCT [35] establishes that salience plays a key role in the first stage (attentional processes) of the observational learning. As this difference in salience might make a difference in the effectiveness of non-branded food placements, the second hypothesis of the present study considers modality as a potential factor that influence children’s food choices:

**H2.** 
*Bimodal as opposed to unimodal food placements will lead to an increase in food choice.*


As far as we know, most of the previous literature about food placements in general (branded and non-branded) has focused on low nutritional value bimodal food placements. Effects have been found when foods belonged to the less healthy category and when they were audiovisually presented [8,9,10,26], whether non-branded [8] or branded [9,10,26]. However, none of the previous literature has used real cartoons to study food placement effects. Our study aimed to extend the knowledge about non-branded foods placed in cartoons by considering two categories that have been scarcely studied in the literature: high nutritional value foods and the unimodal presentation. As previous studies that compare high and low nutritional food placements [8,37] show a higher effect for low nutritional food messages and as studies that compare bimodal and unimodal modality [22,41] show a higher effect for bimodal placements, the third hypothesis considers the interaction between type of food and modality as potential factors that influence food choice.

**H3.** 
*Bimodal low nutritional value as opposed to unimodal low nutritional value: bimodal high nutritional value and unimodal high nutritional value food placements will lead to an increase in food choice.*


#### 1.1.2. Moderating Effects of Children’s Ages

Cartoons are watched daily by children [32]. However, as children grow up, their media preferences change: they become gradually less interested in children’s cartoons targeted at infants and modify their preferences to other media content such as TV series or sitcoms [45]. Older children might see cartoons (especially those targeted to early age audiences) as a type of media aimed at “babies”. Therefore, parasocial relationships with cartoon characters might weaken at a certain age when children stop watching their favorite animated characters [34].

Children’s age has been extensively considered as a moderator variable in the literature about product placement effectiveness [7,10,26,28]. No consistent findings have been found: some studies found significant moderating effects of children’s age [28], but other studies did not [10]. However, as we commented previously, most of the studies used branded food placements, and it might be possible that the results found in branded foods do not apply to non-branded foods. Children’s defenses (associated with age) against the persuasive messages of branded foods might not occur in non-branded food portrayals [46].

In any case, younger children might differ with regard to older children when selecting foods as the latter have a longer learning history and might therefore incorporate more factors to base their preference on [47]. We hypothesize that younger children will be more influenced by food placements not only because they have less established food preferences but also because they might see cartoon characters as objects to later emulate in a greater way than older children. It follows that: 

**H4.** 
*Children’s age will moderate the effects of food placements on food choice regarding the type of food.*


**H5.** 
*Children’s age will moderate the effects of food placements on food choice regarding the modality.*


**H6.** 
*Children’s age will moderate the effects of food placements on food choice regarding the interaction between the type of food and modality.*


## 2. Materials and Methods

### 2.1. Participants

The participants were recruited from a school in Córdoba (Spain). Prior to conducting the experiment, approval was obtained from the Ethics Committee of the University to carry out the study. Information sheets were distributed to the children’s parents, and the signed consent forms were collected. There were 124 children (51.61% girls), and their ages ranged from 7 to 11 years (M = 9.24, SD = 1.19), Table 1.

### 2.2. Procedure

An experimental design was used with two conditions: an experimental condition (visualization of cartoons with food placements) and a control condition (visualization of cartoons without food placements). The participants were assigned to one of the two conditions according to their surname. Each participant in the experimental condition was exposed to 16 scenes of cartoons. Each cartoon scene included a different food placement. Unlike previous experimental studies that manipulate one single food placement per condition, we included an ample sample of foods to analyze how the whole category of foods placed (low and high nutritional value foods) or the modality (unimodal and bimodal) influence food choice. In contrast with previous studies that used animated scenes created exclusively for the experiment, we selected cartoon scenes from fragments of 11 popular real cartoon series: Adventure Time, All Hail King Julien, Fanboy and Chumchum, George of the Jungle, Gravity Falls, Pokemon, Phineas and Ferb, SpongeBob, The Amazing World of Gumball, The Jungle Book and The Ninja Turtles. The decision to use real cartoons was made to improve the external validity—that is, by simulating the natural environment in which children watch cartoons. 

Only in the experimental condition did the scenes contain non-branded food placements. The 16 foods placed were equally grouped into high and low nutritional value foods, following the food criteria of the WHO (2015) [48]. Fast food, sweets, salty snacks, soft and energy drinks were considered as low nutritional value foods, whereas fruits, vegetables, water, cereals, potatoes and pasta were considered as high nutritional value foods. Additionally, food placements were classified in unimodal and bimodal placements, Table 2. 

Scenes of cartoons aimed at children over 7 were randomly selected from the database of a published content analysis [5]—that is, once bimodal low nutritional value food placements were filtered, four scenes were randomly chosen. The same procedure was followed with the other types of food placements. 

### 2.3. Randomization Check

The randomization checks for gender, *χ*
^2^ (1, N = 124) = 0, *p* = 1, and age, *χ*
^2^ (4, N = 124) = 0.288, *p* = 0.991, were successful.

### 2.4. Experimental Stimuli

The 16 food placement scenes were grouped into 4 types. Each type contained 4 scenes. The 4 bimodal low nutritional value foods were portrayed in the following described scenes. In bacon placement scenes (The Amazing World of Gumball), the bacon is portrayed as a character who sings: “Baconman! I am made of bacon; I am the only who can transform a vegetarian in bacon lover because I am made of bacon!” In burrito placement (Fanboy and Chumchum), the main characters are singing a song about working in the freezing-shop and preparing burritos. Fanboy is placed inside a burrito who is moving inside a microwave. In gummy bears placement (Gravity Falls), Lil Gideon uses gummy bears as a strategy to bribe Marbel, and when she sees these sweets, she exclaims, “Gummy bears!” and she starts to eat them. In mayonnaise placement (Adventure Time), Jake (the dog) says to Finn (the human): “I have everything that I could desire: Infinity mayonnaise (and mayonnaise appears), new attractive hairstyle and I have learnt the best magic power: sleep whenever I want”.

The 4 unimodal low nutritional value foods were portrayed in the following described scenes. In cookies placement (SpongeBob), Sandy Cheeks is eating some cookies with Spongebob while they are talking about life in the sea world. In hot dog placement (Fanboy and Chumchum), Fanboy and Chumchum speak like robots. Chumchum asks Fanboy: “Bip, bip, what is this strange object (a fire hydrant)” and Fanboy replies: “bip, bip, I do not know, it seems like a giant hot dog”. In nachos placement (Gravity Falls), Deeper and Soos are celebrating that they have taken pictures of a strange creature. Soos exclaims: “I am going to prepare some victory’s nachos!” and then they high five. In pizza placement (The Amazing World of Gumball), the Gumball family talk to each other while eating pizza.

The 4 bimodal high nutritional value foods were portrayed in the following described scenes. In grapes placement (Adventure Time), Finn and Jake are taking care of a mini robot. Finn says: “Let’s feed the mini robot with these purple things!” Jake interrupt and says: “Do you mean grapes? “And Finn replies: “Yes…whatever these things are” And the mini robot eats the grapes. In lettuce placement (The Ninja Turtles), Splinter master says to Rafael: “Let me tell you a story”, to which Rafael replies: “Sensei, I am not in good mood for stories”. In that moment, Splinter master speaks with Spike (a little turtle) and says: “Spike bits the lettuce if you are in good mood for stories”, and Spike bits the lettuce. In pineapple placement (All Hail King Julien), Julien is looking at the sky, he is worried because he has been arguing with Moris. Suddenly, a shining pineapple character speaks from the night sky and says: It is me, Pineapple! And pineapple character makes Julien reflect about his friendship with Moris. In toast placement (Pokemom), Phanphy is following the smell of something. It is Meowth who is preparing toasts. Meowth says to his friends: “I want to offer you these toasts”.

The 4 unimodal high nutritional value foods were portrayed in the following described scenes. In asparagus placement (Phineas and Ferb), Phineas and Ferb are eating asparagus folks in the kitchen while they are talking. In coconut placement (George of the Jungle), Ursula and Magnolia ask Shep (the elephant), “Are you hungry? Do you want a coconut?” In corn placement (Gravity Falls), Marbel and dipper go to a theme park, they stop in a place where there is a corn stand. Lastly, in water placement (The Jungle Book), Mowgli says to a bird that he is going on an adventure, to which the bird says: Are you prepared? Have you taken enough water?

The animated scenes with food placements had an average duration of 12.63 sec (SD: 5.5 sec), appeared randomly and were presented by inserting a white image between the scenes. The entire video lasted 4 min and 15 sec. The control group watched scenes from the same animated cartoons without food placements.

### 2.5. Measures

After viewing the animated scenes, all children had to complete 16 different decision tasks. Each choice task consisted of selecting, according to their preferences at that time, a food from the four to choose from. Before starting with the choice tasks, instructions were given to children, asking them if they had any doubt. A keyboard was prepared for the experiment and it contained numbers from 1 to 4. Each child had to press the corresponding number in the keyboard according to the question: Which food of the 4 would you like to try at this time?

The software E-prime 2.0 (Psychology Software Tools, Pittsburgh, PA, USA) [49] was used, and it provided the child’s response. The position of the target food was randomly changed in each card to avoid this aspect influencing the choice made.

As shown in Appendix A, food cards contained numbers, words and images. Each food card included 3 novel foods and 1 target food. Images were pictures similar to the target foods portrayed in the cartoon scenes. The foods that appeared on each card belonged mostly to foods of the same category, whether fruits, vegetables or sweets, etc. (Appendix A).

### 2.6. Data Analysis

We conducted a repeated measures ANOVA with food choice as the dependent variable. We inserted food type (low vs. high nutritional value foods) and modality (unimodal vs. bimodal) as within-participant factors. Group (experimental vs. control) and age were inserted as between-participant factors. In cases where Maulchy’s test of sphericity indicated a violation, we applied GreenhouseeGeisser corrections. We tested our hypotheses examining the interaction effects of group with the rest of the factors.

Additionally, in order to test the age differences, we conducted a moderated mediation analysis using Hayes’ PROCESS macro for SPSS [50] (model 1, using 5000 bootstrapping samples). We inserted group as an independent variable and age as moderator variable. To define a threshold for age, we used the Johnson-Neyman technique. Analyses were performed with SPSS statistical software, version 24.

## 3. Results

We expected (H1) that low nutritional value food placements will significantly increase food choice compared to high nutritional value food placements. We did not find main effects of food type on children’s food choices, *F*(1, 120) = 1.877, η_p_^2^ = 0.016, *p*= 0.173. Similarly, no significant effects were found for bimodal placements compared to unimodal placements (H2), *F*(1, 120) = 0.581, η_p_^2^ = 0.005, *p* = 0.448. When considering the interaction effects between food type and modality (H3), no significant effects were found, *F*(1, 120) = 0.248, η_p_^2^ = 0.002, *p* = 0.619. Thus, H1, H2 and H3 were not supported (Table 3).

In contrast, when introducing age as a moderator factor, main effects emerged when considering type of food (H4), *F*(1, 120) = 2.717, η_p_^2^ = 0.087, p = 0.033. However, age did not moderate the effects neither for placement modality (H5), *F*(1, 120) = 1.007, η_p_^2^ = 0.034, p = 0.407 nor for the interaction between food type and modality (H6), *F*(1, 120) = 1.139, η_p_^2^ = 0.038, p = 0.342. Thus, H4 was supported, but H5 and H6 were not supported.

Once we conducted the moderated mediation analysis [50] considering food type (low and high nutritional value food choice), we looked for the definition of Johnson-Neyman age significance region for low nutritional value food choice, which reached a value of 8.95 (age). As indicated in Figure 1, Children under 9 were more likely to choose the low nutritional value foods (compared to high nutritional value foods, Figure 2) placed in the cartoons than the rest of the groups (children in the control group under 9 and children aged over 9, both control and experimental groups).

## 4. Discussion

There is limited knowledge as to whether non-branded food placements can shape children’s food behavior. In this study, we examined the effects of non-branded food portrayals placed in most broadcasted animated cartoons on children’s conative outcomes (choice behavior). We found that low nutritional value food placements were more effective in increasing food choice in younger children (aged 7–8 years) than high nutritional value food placements. Therefore, our results indicate that age moderated the effects of low nutritional value food placements.

Our results might be explained by several reasons. The first reason is described regarding the type of food. Low nutritional value foods are more appealing than high nutritional value foods, not only for children but also for people in general. According to Raghunathan et al. (2006) [51], the taste of low nutritional value foods is inferred to be better than the taste of high nutritional value foods. Additionally, low nutritional value foods are enjoyed more during actual consumption and are preferred in choice tasks [51]. Furthermore, neuroimagery studies, such as one conducted by Meer et al. (2016) [52], found that low nutritional value foods elicit more attention than high nutritional value foods, and for children, they have a higher level of activation in the areas involving reward, motivation and memory while viewing low nutritional value foods. Visual attention studies with eye-tracking cameras [53,54] showed that children pay more attention to low nutritional value foods compared to high nutritional value foods. Implicit affective evaluation studies, such as one by Woodward and Treat (2015) [55], indicated that both added fat and added sugar are associated with more positive affective evaluations. This inherent attraction to low nutritional value foods might be the reason why children are more influenced by these food portrayals placed within media content. Children’s minds might be more activated to messages associated with low nutritional value foods, and thus, children might develop behaviors consistent with these messages.

Second, one possible explanation about the age differences found is that younger children have less established preferences [47]. As children grow up, their dietary habits become more established [56]. That is, the preferences for a certain low nutritional value food may not be as defined for a younger child as they are for an older child; therefore, the youngest children might be more influenced by the appearance of this specific food placement. Additionally, younger children are more impulsive than older children in go/no-go tasks [57], which, in the present study, might mean that this age group has less self-control to refrain their impulses of selecting the placed foods. Although middle childhood comprises primary school—that is, children from 6 to 12 years of age—some organizations such as the Common Sense Media Organization [58] or centers such as the Centers for Disease Control and Prevention [59] establish differences in this age range with regard to children’s cognitive, social and emotional development. In the first stage of middle childhood (6–8 years of age), children are less aware of the point of view of others than in the second stage (9–11 years of age). As cartoon characters might be seen as models with other points of view, it might not be strange that younger children are less aware of these perceptions and, therefore, are more influenced by the food portrayals. Additionally, the age differences that we found might be because as children grow up, they become less interested in cartoons. As public data shows, younger children watch cartoons the most, and older children begin to change their interests to other forms of leisure media [60]. Therefore, younger children might be more open and receptive to food messages in cartoons, whereas older children might be more disconnected from these messages.

Third, it is important to highlight that the food placement scenes selected for the present study were randomly chosen from the database of a content analysis that examined the presentation of non-branded foods placed in animated cartoons targeted to children over 7 [5]. In that content analysis [5], traditional execution factors were considered named as factors of prominence: placement duration (in seconds), screen position (background vs. foreground), plot connection (low vs. high) and modality (unimodal vs. bimodal). It was found that low nutritional value foods were more prominent than high nutritional value foods when it comes to animated cartoons targeted to audiences over age 7. Similarly, in Matthes and Naderer (2018) [4]’s content analysis of foods placed in children’s movies which also measured placement interaction and placement consumption, higher persuasive potential of low nutritional value food placements compared to high nutritional value food placements was found. According to SCT [35], both the salience (prominence) and the context (positive, neutral and negative) of messages placed in media influence modelled behaviors. As low nutritional value food placements are more prominent and present more positive contexts (celebration, joy, pleasure) than high nutritional value food placements, our results might be also explained by the different salience and enthusiasm between both types of foods.

The present study has some limitations. First, we exposed children to placements in excerpts of the cartoon series. Although foods in cartoons are prevalent, it is not an easy task to find an episode that contains several types of foods (low nutritional value foods and high nutritional value foods) combined with different types of placement; therefore, we used scenes instead of using full episodes to control these variables. However, the length of the cartoon video should be tested in future research. Second, although food cards were designed to include foods that belonged to the same food category (fruits, vegetables, snacks, sweets, etc.), it was not easy to control the degree of appeal of the foods represented on each card. This could have affected the measurement of non-branded food placement effects, as both the experimental and control groups chose the target food at a high rate. We suggest taking this aspect is taken into account when designing options for a task that involves food choice. Third, in the present study, modality did not play a key role in food placement effectiveness. As aforementioned, modality is among the most studied execution factors in the literature about product placements [22,41,43]. However, recent studies have explored other variables that might be more strongly implicated in the effectiveness of food placements. For example, Binder, Naderer and Matthes (2019) [38] found that children’s choices were affected by small variations in endorsement—that is, in function of the number of animated characters eating the food. Further, Naderer, Matthes and Zeller (2017) [26] explored how the different levels of interaction between characters and products (foods) influence children’s food choices. In the present study, we controlled type of food and modality. However, we did not control other variables that might affect children’s food choices.

Our findings concerning the behavioral effects of non-branded food placements have considerable practical implications. Children spend many hours watching cartoons that include food placements. According to a recent content analysis of non-branded food placements in cartoons, children are exposed, on average, to one placement less than every 5 min [5]. Further, when it comes to cartoons targeted to children over 7, a high percentage of food messages (approximately 60%) are non-supportive to educational food content [5]. That is, not only are low nutritional value food placements prevalent, but also these portrayals appear associated with positive contexts. In our study, low nutritional value foods placed in cartoons seem to have a greater short-term impact than high nutritional value foods for young children. Combining both types of studies (content analysis and the present experiment), the consequences of portraying low nutritional food placements in cartoons targeted to children over 7 are negative when it comes to educate recommended eating habits. With regard to high nutritional value food placements, although we did not find short-term effects on this type of food, it is necessary to investigate whether long-term effects could take place on these foods.

## 5. Conclusions

We hope that our present study paves the way for future research. Considering the potential influence on children’s diet, these results should not be overlooked. The type of food plays a key role in non-branded food placement effectiveness, and unfortunately, the effects are not aligned with the WHO’s (2015) [48] food consumption recommendations. The problem of childhood obesity requires that urgent strategic measures be taken related not only to advertisements and packaging but also to other strategies, such as food placements.

Young children constitute a sector of the public that should be among the main foci of political strategic measures. The younger the children are, the more moldable their behavior is, both in a positive [61] or in a negative way. We recommend policy makers to take these results into account when designing strategies to prevent obesity, specifically when cartoons are targeting the youngest children.

## Figures and Tables

**Figure 1 ijerph-16-05032-f001:**
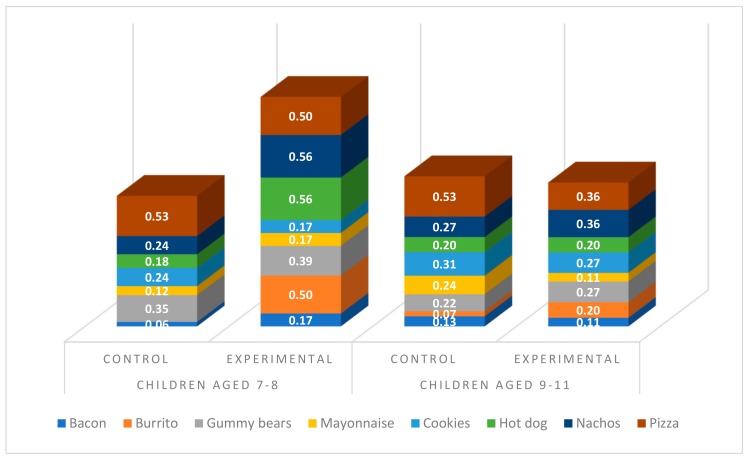
Low nutritional value foods chosen (%) by children according to condition and age group.

**Figure 2 ijerph-16-05032-f002:**
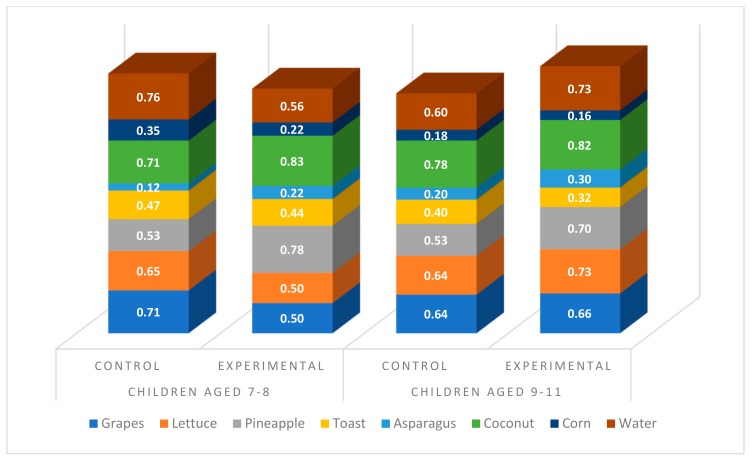
High nutritional value foods chosen (%) by children according to condition and age group.

**Table 1 ijerph-16-05032-t001:** Age distribution of participants.

Age, y	Total Sample (N = 124), N (%)	Control Group (N = 62), N (%)	Experimental Group (N = 62), N (%)
7	11 (8.9%)	6 (9.7%)	5 (8.1%)
8	24 (19.4%)	11 (17.7%)	13 821%)
9	33 (26.6%)	17 (27.4%)	16 (25.8%)
10	36 (29%)	18 (29%)	18 (29%)
11	20 (16.1%)	10 (16.1%)	10 (16.1%)

**Table 2 ijerph-16-05032-t002:** Placed foods according to food type and modality

	Bimodal	Unimodal
Low nutritional value foods	Bacon	Cookies
	Burrito	Hot dog
	Gummy bears	Nachos
	Mayonnaise	Pizza
High nutritional value foods	Grapes	Asparagus
	Lettuce	Coconut
	Pineapple	Corn
	Toast	Water

**Table 3 ijerph-16-05032-t003:** Interaction effects of food placements explaining food choice.

Between Subjects	df	*F*	η_p_^2^	*p*
Group × Food type	1	1.877	0.016	0.173
Group × Modality	1	0.581	0.005	0.448
Group × Food type × Modality	1	0.248	0.002	0.619
Group × Food type × Age	1	2.717	0.087	0.033
Group × Modality × Age	1	1.007	0.034	0.407
Group × Food type × Modality × Age	1	1.139	0.038	0.342

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
