# Peer review of "Investigating the Effects of Non-Branded Foods Placed in Cartoons on Children’s Food Choices through Type of Food, Modality and Age"

_ijerph, 2019, doi:10.3390/ijerph16245032_

Round 1

Reviewer 1 Report

This is a very interesting paper, that could provide significant contribution to our understanding of non-branded product placement effects on children’s food choices.

This paper presents an experiment where children were exposed to cartoon scenes with non-branded food, and subsequently completed a series of choice tasks where they indicated their preference for foods that were and were not shown in the cartoons. The experiment included a control condition of children who were exposed to scenes from the same cartoons that did not included food placements. The foods in the cartoon scenes varied on two factors: whether they were nutritionally recommended foods (like grapes, asparagus, and lettuce) or nutritionally less recommended foods (like bacon, cookies, and burritos); and whether they were placed bimodally (both visual and verbal) or unimodally (either visual or verbal). The results indicate that children were more likely to choose less recommended foods were placed bimodally compared to the control children, but that they were no more likely to choose highly recommended foods placed bimodally or less recommended foods placed unimodally. The paper is well-written and interesting.

Comments:

In section 1.1.1, you distinguish food placements as less recommended vs highly recommended. Could you define these? On first reading, I interpreted it to mean foods that are commonly recommended to children (for example, baby carrots) versus foods that aren’t often recommended to children (for example, brussels sprouts). But when I looked at the articles you reference, I couldn’t find a common definition. The first article that used raspberries (Binder, Naderer & Matthes 2019) describe a manipulation where the media shows either few children eating the food, or many children eating the food, while the second article that used mandarins (Naderer et al 2018) describe a manipulation where the food is either of high nutritional value (mandarins) or low nutritional value (fruit snacks). I think adding a definition would clarify your meaning and discussion of the literature.

Your second hypothesis, section 1.1.2, seems to be missing something. As written, it sounds like it is measuring the same things as hypothesis 1, but with only a younger subsample of your participants. If you are looking at the moderating effect of age, as you indicated, should the hypothesis not be that younger children will be more likely to choose the placed foods compared to older children in the experimental condition?

Study methods: Were the scenes chosen pretested for how highly recommended the food is by the characters in the scene? You describe 2 examples of scenes used, and it seems like the one from Gravity Falls that had bimodal placement of a less recommended food involves much more enthusiasm for the food on the part of the characters than the scene from George of the Jungle that had a unimodal placement of a highly recommended food. Binder, Naderer & Matthes (2019) suggest that children’s choices are affected by small variations in endorsement, such as using the number of characters eating the food as evidence of how many children like the food. Which brings to mind the question of whether the scenes were pretested so that the highly recommended compared to less recommended and bimodal compared to unimodal presentation scenes all had equivalent endorsement from the characters. Could an alternate explanation of your results be that when a food is highly recommended or when a food is presented unimodally, the characters in the scene present less enthusiasm for it, thus children are less likely to choose it?

Data Analysis: Since this is a repeated-measures design, I’m not sure I understand your choice to analyze the data with a univariate anova instead of using a repeated-measures anova. For H2, is the significant group*age interaction indicating that younger children are driving the effect in H1a? In other words, that younger children preferentially choose less recommended foods place bimodally when compared to all other groups (older children in both conditions, as well as younger children in the control group)? I think providing a table that compares choice across the 4 groups would help clarify this result. Currently, you provide only the percent chosen across the 2 younger groups, but not the older groups. With only the younger group, there is no way to conclude that there is an effect of age.

Overall:

I think this is a very interesting paper with a well-designed study. I think, however, that it needs to be reanalyzed using a method that can capture repeated measures, and hypothesis 2 should be rewritten and analyzed such that it captures differences between the younger and older children.

Reviewer 2 Report

Thank you for this investigation into the extent to which non-branded food placement into children’s cartoons affects their subsequent food choice. Overall this appears to be a useful contribution to the field but requires some detailed explanation of various aspects of the methodology and analyses.

Rationale

Proposed cognitive mechanisms need to be more clearly described. E.g., 74-76 Explain relationship between attributes, benefits and perceptions and the argument (with evidence) that you are making about this relationship. 90-100 Discussion of SCT needs to be clarified, and especially connected to your hypotheses. E.g., 103 argues that behavior is first step toward consumption – evidence for this?

Discussion at 151-153 seems to suggest that your study has already been done. Clarify what these 4 studies did and did not find and what the present study does differently.

Wording at 156-163 is a bit confusing regarding hypotheses. Is there one hypothesis or three? (the wording in the Results section is much clearer, referring to these hypotheses as 1a, 1b, 1c). Why is the last hypothesis singled out as a Research Question rather than a hypothesis? All the variables are the same as in the hypotheses – why not just word it and test it as a hypothesis? Current wording doesn’t suggest directionality, but rationale would suggest you are hypothesizing: “Children who see the foods in cartoons will be more likely to choose the placed foods compared to children who did not see the foods in cartoons in regard to: (d) highly recommended foods placed unimodally”

Methodology

Section 2.2 Design not clearly explained. How, exactly, do type of food, healthfulness of food, and number of modalities vary between the 16 slides? You have this explained more clearly in section 2.4 so maybe lines 211-214 should be removed from Procedure section, leaving description of the scenes to the Stimuli section.

209-210 Using surnames to assign children to conditions is not random assignment.

Please describe the food cards. 215 Do they contain words or images? If images, are these taken directly from the cartoon, or are they a different image of the food depicted in the cartoon?

Please clarify the procedure. (248-250) Were the children asked questions orally or by a computer? Explain how ePrime was used to record data (i.e., do children touch a screen or press a key to indicate their preference; or do they respond orally?).

How are foods identified as healthful or unhealthful? For example, what makes a burrito (which contains protein and vegetables) less healthful than toast (an empty, refined carb)?

In choice task, are 3 of the foods novel and the 4th a target food from the cartoon? Please clarify this.

Analysis/Results

Design, analysis, and comparisons are all a bit unclear. In several places, main effects are claimed only at certain levels of your IVs (e.g., 258; 263; 287), which precludes them from being main effects. Interactions are not discussed.  271 and 273 unclear what two groups the differences are between.   Are two age groups compared? If so, what is the second age group (10-11? 9-11?). Or, did you simply re-do the same analyses but only with the 7-8-year-olds?

Did you try analyzing each food individually? Your one figure suggests you have an effect for gummy bears but not for mayonnaise. This would be worth talking about. Perhaps some of your healthful foods would also show a difference if analyzed individually.

Tables are confusing due to use of “fixed factors” and “hypothesis” columns. Unclear what “group” and “age” mean, especially if there was no age comparison.

Perhaps under Limitations or around 360, discuss how many milliseconds the foods appear on the screen and how this varies between foods.

A few minor points of grammar (e.g., medium vs. media; and others); word choice (e.g., 183, 187-188; 249 “provided” vs “recorded”; 291 “effects of” vs “effects on”); & passive voice (e.g., 123)

Round 2

Reviewer 1 Report

This revised manuscript is much improved. I appreciate the detailed potential explanations that were included in the Discussion section and the figures showing choice shares of low nutritional value foods and high nutritional value foods. I think this paper is interesting and the results are important from a public health perspective.